# Meru couples planar cell polarity with apical-basal polarity during asymmetric cell division

Jennifer J Banerjee[1], Birgit L Aerne[1], Maxine V Holder[1], Simon Hauri[2,3†], Matthias Gstaiger[2,3], Nicolas Tapon[1]*

[1]Apoptosis and Proliferation Control Laboratory, The Francis Crick Institute, London, United Kingdom; [2]Department of Biology, Institute of Molecular Systems Biology, ETH Zürich, Zürich, Switzerland; [3]Competence Center Personalized Medicine UZH/ETH, Zürich, Switzerland

**Abstract** Polarity is a shared feature of most cells. In epithelia, apical-basal polarity often coexists, and sometimes intersects with planar cell polarity (PCP), which orients cells in the epithelial plane. From a limited set of core building blocks (e.g. the Par complexes for apical-basal polarity and the Frizzled/Dishevelled complex for PCP), a diverse array of polarized cells and tissues are generated. This suggests the existence of little-studied tissue-specific factors that rewire the core polarity modules to the appropriate conformation. In *Drosophila* sensory organ precursors (SOPs), the core PCP components initiate the planar polarization of apical-basal determinants, ensuring asymmetric division into daughter cells of different fates. We show that Meru, a RASSF9/RASSF10 homologue, is expressed specifically in SOPs, recruited to the posterior cortex by Frizzled/Dishevelled, and in turn polarizes the apical-basal polarity factor Bazooka (Par3). Thus, Meru belongs to a class of proteins that act cell/tissue-specifically to remodel the core polarity machinery.

*For correspondence: nic.tapon@crick.ac.uk

Present address: †Department of Clinical Sciences, Lund University, Lund, Sweden

Competing interests: The authors declare that no competing interests exist.

## Introduction

Polarity is a fundamental feature of most cells and tissues. It is evident both at the level of individual cells (e.g. apical-basal polarity in epithelia, budding in *Saccharomyces cerevisiae* [*St Johnston and Ahringer, 2010*, *Martin and Arkowitz, 2014*]) and groups of cells (e.g. planar cell polarity (PCP) in epithelia [*Singh and Mlodzik, 2012*; *Devenport, 2014*]). However, despite the fact that different cell types use a common set of molecules to establish and maintain polarity (Par complexes, Fz-PCP pathway), the organization of polarized cells and cell assemblies varies dramatically across different species and tissues (*Bryant and Mostov, 2008*). This implies the existence of factors that act in a cell or tissue-specific manner to modulate/rewire the core polarity machinery into the appropriate organization. Despite many advances in our understanding of polarity in unicellular and multicellular contexts, little is known about the identity or function of such factors.

An example of polarity remodeling is the process of asymmetric cell division (ACD), where cells need to rearrange their polarity determinants into a machinery capable of asymmetrically segregating cell fate determinants, vesicles and organelles, as well as controlling the orientation of the mitotic spindle. ACDs result in two daughter cells of different fates and occur in numerous cell types and across species. Well-studied examples include budding in *Saccharomyces cerevisiae*, ACD in the early embryo of *Caenorhabditis elegans*, or ACD of progenitor cells in the mammalian stratified epidermis and neural stem cells in the mammalian neocortex (reviewed in [*Dewey et al., 2015*; *Griffin, 2015*; *Inaba and Yamashita, 2012*]). In *Drosophila melanogaster*, the study of germline stem

cells, neuroblasts (neural stem cells) and sensory organ precursors (SOPs) has greatly contributed to our understanding of the cell biology and molecular mechanisms of ACD (*Knoblich, 2008*; *Schweisguth, 2015*; *Spradling et al., 2011*).

SOPs (or pI cells) divide asymmetrically within the plane of the epithelium into pIIa and pIIb daughter cells. pIIa and pIIb themselves divide asymmetrically to give rise to the different cell types of the external sensory organs (bristles), which are part of the peripheral nervous system and allow the adult fly to sense mechanical or chemical stimuli (*Roegiers et al., 2001b*; *Hartenstein and Posakony, 1989*; *Gho et al., 1999*; *Jarman, 2002*; *Stocker, 1994*). Individual SOPs are selected by Notch-dependent lateral inhibition from multicellular clusters of epithelial cells expressing proneural genes (proneural clusters) (*Reeves and Posakony, 2005*; *Cubas et al., 1991*; *Skeath and Carroll, 1991*; *Hartenstein and Posakony, 1990*; *Simpson, 1990*).

The unequal segregation of cell fate determinants (the Notch pathway modulators Numb and Neuralized), which specifies the different fates of the daughter cells, requires their asymmetric localization on one side of the cell cortex prior to mitosis (*Le Borgne and Schweisguth, 2003*, *Rhyu et al., 1994*). This is achieved by remodeling the PCP and apical-basal polarity systems in the SOP, and by orienting the spindle relative to the tissue axis (*Schweisguth, 2015*; *Gho and Schweisguth, 1998*). The epithelial sheet that forms the pupal notum (dorsal thorax), where the best-studied SOPs are located, is planar polarized along the anterior-posterior tissue axis, with the transmembrane receptor Frizzled (Fz) and its effector Dishevelled (Dsh) localizing to the posterior side of the cell cortex, while the transmembrane protein Van Gogh (Vang, also known as Strabismus) and its interactor Prickle (Pk) are found anteriorly (*Bellaïche et al., 2004*; *Ségalen et al., 2010*). The apical-basal polarity determinants central to SOP polarity are the PDZ domain-containing scaffold protein Bazooka (Baz, or Par3), atypical Protein Kinase C (aPKC) and Partitioning defective 6 (Par6), which localize apically in epithelial cells and the basolaterally localized membrane-associated guanylate kinase homologues (MAGUK) protein Discs-large (Dlg) (*St Johnston and Ahringer, 2010*). In most epithelial cells, these proteins localize uniformly around the cell cortex, whereas in SOPs they show a striking asymmetric localization during mitosis: the Baz-aPKC-Par6 complex is found at the posterior cell cortex, opposite an anterior complex consisting of Dlg, Partner of Inscuteable (Pins) and the G-protein subunit $G\alpha_i$ (*Schaefer et al., 2001*; *Bellaïche et al., 2001b*). The Fz-Dsh complex provides the spatial information for the Baz-aPKC-Par6 complex, while Vang-Pk positions the Dlg-Pins-$G\alpha_i$ complex (likely through direct interaction between Vang and Dlg) (*Bellaïche et al., 2004*; *Besson et al., 2015*; *Bellaïche et al., 2001b*). The asymmetric distribution of the polarity determinants then directs the positioning of cell fate determinants at the anterior cell cortex (*Bellaïche et al., 2004*, *2001a*, *2001b*; *Wirtz-Peitz et al., 2008*). Additionally, Fz-Dsh and Pins orient the spindle along the anterior-posterior axis by anchoring it on both sides of the cell via Mushroom body defective (Mud, mammalian NuMA) and Dynein (*Ségalen et al., 2010*; *Bowman et al., 2006*).

The planar symmetry of the Baz-aPKC-Par6 complex in SOPs is initially broken in interphase via Fz-Dsh, and is independent of the Dlg-Pins-$G\alpha_i$ complex (*Besson et al., 2015*). Once this initial asymmetry is established, the core PCP components become dispensable for Par complex polarization at metaphase due to the mutual antagonism between the opposing polarity complexes, which then maintains asymmetry during cell division (*Bellaïche et al., 2004*, *2001b*; *Besson et al., 2015*). Indeed, Baz is still polarized in *fz* mutants during mitosis, but losing both *pins* and *fz* results in Baz spreading uniformly around the cortex (*Bellaïche et al., 2001b*). Crucially, it is unclear how Fz-Dsh can transmit planar information to the Baz-aPKC-Par6 complex in SOPs but not in neighboring epithelial cells. The cell-type dependent coupling between PCP and apical-basal polarity suggests the involvement of unknown SOP-specific factors in this process (*Besson et al., 2015*; *Schweisguth, 2015*).

The four N-terminal RASSFs (Ras association domain family) in humans (RASSF7-10) have been associated with various forms of cancer, but the exact processes in which these scaffolding proteins act remain mostly elusive (reviewed in [*Sherwood et al., 2009*]). *Drosophila* RASSF8, the homologue of human RASSF7 and RASSF8, is required for junctional integrity via Baz (*Zaessinger et al., 2015*; *Langton et al., 2009*). Interestingly, human RASSF9 and RASSF10 were found in an interaction network with Par3 (the mammalian Baz homologue) and with several PCP proteins (*Hauri et al., 2013*). The *Drosophila* genes *CG13875* and *CG32150* are believed to be homologues of human *RASSF9*

and *RASSF10,* respectively (*Sherwood et al., 2009*) and remarkably, *CG32150* mRNA is highly enriched in SOPs (*Reeves and Posakony, 2005*; *Buffin and Gho, 2010*).

Here, we show that Meru, encoded by *CG32150,* is an SOP-specific factor, capable of linking PCP and apical-basal polarity. Meru localizes asymmetrically in SOPs based on the polarity information provided by Fz/Dsh, and is able to recruit Baz to the posterior cortex.

## Results

## Meru localizes asymmetrically in SOPs and its loss causes bristle defects

In order to study the function of *Drosophila CG32150,* a homologue of human RASSF9 and RASSF10 (*Sherwood et al., 2009*), we generated an antibody to reveal its endogenous localization. *CG32150* hereafter will be referred to as *meru* (the Bengali word for 'pole'), owing to its polarized localization. In third instar wing imaginal discs (the larval precursors to the adult wings), Meru was only detected in SOP cells (*Figure 1A–B'* and *Figure 1—figure supplement 1F–F''*), which can be identified by the expression of Hindsight (Hnt) (*Figure 1—figure supplement 1F–F''*), a marker for specified SOP cells (*Koelzer and Klein, 2003*). This is consistent with previous reports indicating that *meru* mRNA is highly expressed in SOPs, as it is a transcriptional target of the proneural transcription factors of the Achaete-Scute complex (AS-C) (*Reeves and Posakony, 2005*; *Buffin and Gho, 2010*). Strikingly, Meru was highly enriched to one side of the cell cortex (*Figure 1B–B'*). Asymmetric membrane localization of Meru could also be observed in interphase SOPs of the pupal notum using a CRISPR-mediated N-terminal GFP-tagged knock-in (*Figure 1C–C'*). Approximately 15–16 hr after puparium formation (APF), notal SOPs (pI, which give rise to the thoracic microchaetes) start to divide along the anterior-posterior axis into pIIa and pIIb (*Figure 1C''–C'''*). Live imaging revealed that GFP-Meru was localized to the apical membrane, and was asymmetrically enriched at the posterior cortex prior to mitosis (*Figure 1D–D'*). Interestingly, just before the onset of mitosis, GFP-Meru spread around the cell cortex (both apically and basolaterally) and was partitioned into both daughter cells, although it remained enriched at the posterior side (*Figure 1D–D'*).

Using CRISPR, we created two different deletion mutants for *meru: meru[1]* harbors a deletion of 1585 bp and an insertion of 3 bp, while *meru[2]* harbors an 839 bp excision, removing exon 1 (*Figure 1—figure supplement 1A–C*). Using the *meru[1]* allele we were able to confirm that the signal detected by the Meru antibody was specific, both in whole animals (*Figure 1—figure supplement 1F–G''*) and in *hsFLP*-induced mitotic clones of *meru[1]* (*Figure 1—figure supplement 1H–H'*).

Both *meru* alleles were viable and could be kept as healthy homozygous stocks. Homozygous mutant animals or heterozygotes in trans over a deficiency line displayed sensory bristle defects, such as duplicated bristles at the wing margin, missing or additional macrochaetes on the thorax, and split bristles on the thorax and abdomen (*Figure 1E–H''*). Quantification of the two most frequent defects - the duplicated stout bristle defect of the anterior wing margin (*Figure 1I* and outlined in *Figure 1—figure supplement 1D*) and the thoracic missing macrochaete defect (*Figure 1I'* and *Figure 1—figure supplement 1E*) suggests that the *meru[2]* allele is a hypomorph since both defects were weaker than in *meru[1]* mutants and slightly increased when crossed to a deficiency line (*Df(3L)BSC575*) (*Figure 1I–I'* and statistics in *Supplementary file 1*). Indeed, an in-frame translation start site present in exon 2 is predicted to encode a protein lacking amino acids 1–129. This would only truncate the first five (out of 106) amino acids of the conserved RA (Ras association) domain. Therefore the *meru[1]* allele was used for subsequent experiments. To confirm that the bristle defect was caused by *meru* loss of function, we expressed GFP-tagged Meru under the control of the *Ubiquitin-63E* promoter (*ubi-GFP-meru*) in *meru[1]* mutants. Indeed, expression of *ubi-GFP-meru* almost completely abolished the stout bristle and macrochaete defects when compared to the *ubi-GFP* control (*Figure 1I–I'* and *Supplementary file 1*). The quantification of the missing macrochaetes also allowed us to rule out a connection between Meru and the prepatterning genes (e.g. Wingless, Iroquois and Pannier) (reviewed in [*Simpson, 2007*]), as the missing macrochaetes in *meru[1]* mutants did not follow a specific pattern and almost all positions were affected (*Figure 1—figure supplement 1E*).

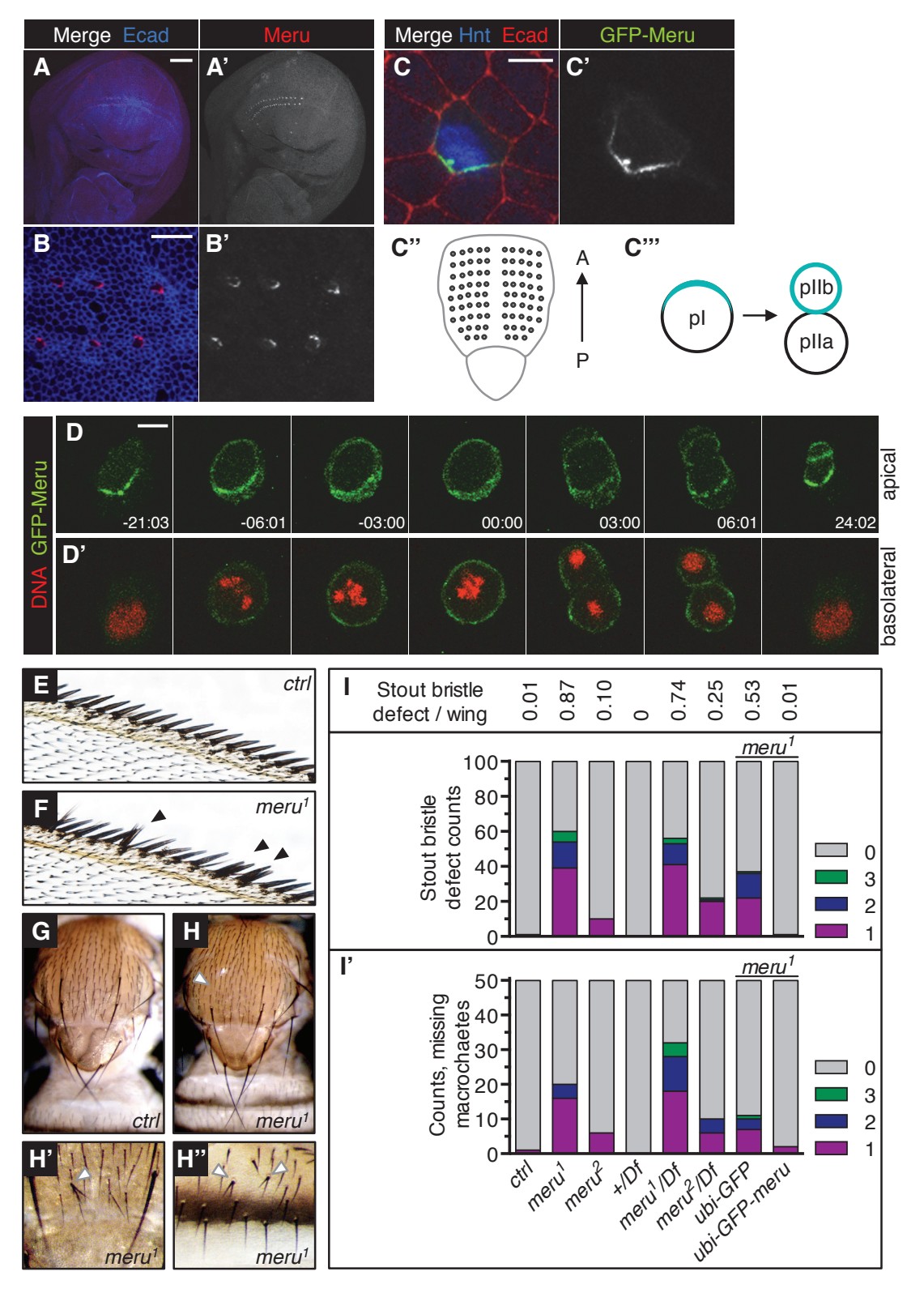

**Figure 1.** Meru localizes asymmetrically in SOP cells and loss of *meru* causes bristle defects in adult flies. (**A–A'**) Third instar wing imaginal disc stained for Meru (red) and E-cadherin (blue). Meru localizes exclusively in SOP cells. Scale bar = 50 µm. (**B–B'**) Higher magnification view of the same disc showing the presumptive wing margin region. Meru localizes asymmetrically to one side of the cell cortex in SOP cells. Scale bar = 10 µm. (**C–C'**) Pupal notum at 15 hr APF stained for E-cadherin (red), Hindsight (blue) and expressing the GFP-Meru knock-in (green). Scale bar = 5 µm. In this and all

*Figure 1 continued on next page*

*Figure 1 continued*

subsequent notal images, anterior is to the top. (C″) Diagram of the pupal notum showing the distribution of the SOP cells (grey circles). (C‴) Diagram outlining the asymmetric division of SOPs (pI), which divide along the anterior-posterior axis into two daughter cells (pIIa and pIIb) of different fates (cell fate determinants are highlighted in turquoise). (D–D′) Time-lapse of GFP-Meru (green) and DNA (red, (D′) localization in a dividing SOP cell. Apical (D) and basolateral projections (D′) are shown for each time point. Time is given in (min:sec) relative to the onset of mitosis (00:00). Scale bar = 5 µm. (E–H″) Images of adult *meru¹* mutant (F, H–H″) and wild type control flies (E, G). *meru¹* mutants have duplicated stout bristles on the wing margin (F), missing thoracic macrochaetes (H), split thoracic (H′) and abdominal (H″) microchaetes (defects are indicated by white and black arrows). (I) Quantification of duplicated stout bristles for the indicated genotype (n = 100). The average duplication per wing is written above the chart. (I′) Quantification of missing thoracic macrochaetes (n = 50). (I–I′) Bristle defects in different *meru* mutants (*meru¹* and *meru²*; *Df: Df(3L)BSC575,* deficiency line for *meru*). Expression of *ubi-GFP-meru* rescues the bristle defects of *meru¹*. See **Supplementary file 1** for statistics.

The following source data and figure supplement are available for figure 1:

**Source data 1.** Source data *Figure 1*.
**Figure supplement 1.** Disruption of *meru* using the CRISPR/Cas9 system.

## Meru co-localizes and interacts with Baz, Dsh and Fz

As Meru was asymmetrically localized in SOPs, we wished to test its position relative to the different ACD polarity determinants (illustrated in *Figure 2—figure supplement 1A*). Endogenous GFP-Meru co-localized with Baz and Dsh in interphase SOPs of the pupal notum (*Figure 2A–B″*). Meru stainings in third instar wing imaginal discs also showed its co-localization in interphase SOPs with Baz, Fz and Dsh (*Figure 2—figure supplement 1B–D″*), but not with Vang (*Figure 2—figure supplement 1E–E″*). In premitotic SOPs, GFP-Meru localized opposite Pins (*Figure 2C–C″*).

Since Meru co-localized with Baz and Dsh in SOPs, and previous data indicated an association of mammalian RASSF9/10 with the mammalian homologues of these proteins (Par3 and Dvl1-3) (*Hauri et al., 2013*), we tested whether this physical association was conserved for the *Drosophila* proteins. Indeed, co-immunoprecipitation experiments in S2 cells showed that Meru associates with Baz (*Figure 2D*) and Dsh (*Figure 2E*). We also found that Meru co-immunoprecipitated with Fz (*Figure 2F*) even after Dsh depletion (*Figure 2—figure supplement 1F*). It is likely that the interaction between Meru and Dsh is direct, since it was also identified in a high-throughput yeast two-hybrid dataset (*Formstecher et al., 2005*). The last four C-terminal amino acids of Meru resemble a PDZ binding motif (PBM), which we hypothesized could bind the PDZ domains in Baz and/or Dsh. Indeed, deletion of the PBM (ΔPBM) strongly disrupted the interaction with Baz (*Figure 2D* and *Figure 2—figure supplement 1G*). In contrast, loss of this motif did not affect binding to Dsh or to Fz (*Figure 2—figure supplement 1G*). Thus, Meru can interact with both the PCP proteins Fz and Dsh and the apical-basal polarity determinant Baz.

Dsh and Fz together establish the initial asymmetry in SOPs and are required for the asymmetric enrichment of Baz (*Besson et al., 2015*). In order to identify where Meru can be placed in this hierarchy, we examined the localization of ubiquitously expressed GFP-Meru in mitotic mutant clones of *fz*, *dsh* and *baz*. GFP-Meru localization at the apical membrane required both Fz and Dsh (*Figure 3A–B″*, *Figure 3—figure supplement 1A–B″*), but not Baz (*Figure 3C–C″*, *Figure 3—figure supplement 1C–C″*). Thus, Fz/Dsh recruit Meru to the apical cell cortex where it co-localizes with the Baz-aPKC-Par6 group opposite Vang-Pk and Dlg-Pins-Gα$_i$ complexes.

## Genetic interactions between ACD polarity components and *meru*

The relatively low penetrance of the sensory bristle phenotypes observed in *meru* mutants (*Figure 1*) is unsurprising given the high level of redundancy amongst ACD machinery components. For instance, neither *fz* nor *pins* loss is sufficient to disrupt Baz polarization, while both mutants in combination cause loss of Baz polarity (*Bellaïche et al., 2001b*). To test this redundancy, we perturbed the expression of several ACD polarity components in a *meru* mutant background and used the duplicated stout bristle defect at the wing margin as a readout. RNAi knockdown of *baz*, *pins* and *Gα$_i$* in the wing caused duplicated stout bristle defects similar to *meru* loss of function (albeit to varying degrees) (*Figure 3D,E,I*) (*Kopein and Katanaev, 2009*). Expression of these RNAi lines in *meru¹* mutants significantly increased the stout bristle defects, with the most severe effect seen for the *baz*

knockdown, in which many of the bristles were lost altogether (*Figure 3D–I* and *Supplementary file 1*). Indeed, *baz* mutant clones in the notum also show complete loss of microchaetes (*Roegiers et al., 2001a*).

Flies harboring the *dsh*[1] allele, a PCP-specific allele (*Boutros et al., 1998*), can be kept as a homozygous stock. Both the duplicated stout bristle and the missing macrochaete defects of *meru*[1] were strongly enhanced in combination with *dsh*[1] (*Figure 3—figure supplement 1D–D'* and *Supplementary file 1*). Similarly, both *fz* RNAi and misexpression markedly worsened the stout bristle defect of *meru*[1] flies, while activation of Wg signaling via ectopic *armadillo (arm)* expression had no significant effect (*Figure 3—figure supplement 1E* and *Supplementary file 1*). Thus, Meru plays an important role in SOP ACDs, which can be revealed by reducing the function of other polarity components.

## Loss of *meru* causes asymmetric cell division defects

To test whether SOP asymmetric cell divisions were impaired in *meru* mutants, we imaged Pon-GFP (GFP-tagged Partner of Numb) expressed in SOPs under the control of the *neuralized-Gal4* driver in the pupal notum. In control animals, Pon-GFP becomes asymmetrically enriched at the anterior cortex of the SOP cell before mitosis onset (*Figure 4A*) (*Roegiers et al., 2001b*; *Bellaïche et al., 2001a*). It is then partitioned into the pIIb cell during the division, which occurs parallel to the anterior-posterior axis. In contrast, in *meru*[1] mutants, the Pon-GFP crescent was frequently not aligned along the anterior-posterior axis (*Figure 4B*), the division axis was misaligned with the Pon-GFP crescent (*Figure 4B–B'*) or the crescent appeared broader (*Figure 4B''*), all ultimately resulting in mis-segregation of Pon-GFP into both daughter cells.

To quantify these defects, we first calculated the polarization coefficient for Pon-GFP (the polarization coefficient will approach 0 the more uniform the distribution). In *meru*[1] mutants, the Pon-GFP crescent was significantly less polarized compared to control pupae (*Figure 4C*). Secondly, we quantified the Pon-GFP mis-segregation by calculating the ratio of Pon-GFP signal intensity of the two pII cells after division (a ratio of one means equal segregation) and found that the intensity ratios were significantly increased in *meru*[1] mutants compared to control (*Figure 4D*). Finally, the division angle alignment relative to the anterior-posterior body axis was less accurate in *meru*[1] mutants, as the standard deviation (SD) of the division angles was almost double in mutants (SD = 68.33) compared to control pupae (SD = 35.44) (*Figure 4E*). The polarization defect and the mis-segregation of Pon in *meru* mutants were similar to, albeit weaker than, phenotypes described for *baz* and *pins* mutants, where either a weak Pon/Numb crescent is present or Pon/Numb are uniformly spread around the cell cortex (*Roegiers et al., 2001a*; *Bellaïche et al., 2001b*). However, the randomization of the division angle relative to the anterior-posterior axis resembled loss of *fz* (*Bellaïche et al., 2001a*, *2004*) or *dsh* (*Bellaïche et al., 2004*; *Gomes et al., 2009*) function, as these orient the spindle (via Mud) along the anterior-posterior axis (*Ségalen et al., 2010*). Spindle orientation along the anterior-posterior axis is not affected in *pins* or *baz* mutants (*Bellaïche et al., 2001b*).

## Baz polarization is impaired in *meru* mutants

Meru is required for correct ACDs in SOPs - but what is its molecular function? Fz/Dsh have recently been reported to be responsible for the asymmetric enrichment (planar polarization) of the Baz-aPKC-Par6 complex in SOPs prior to mitosis (*Besson et al., 2015*). Since Meru was expressed in SOPs under the control of the proneural factors and interacted with both Fz/Dsh and Baz (*Figure 2D–F*), we hypothesized that Meru could be an SOP-specific factor linking Fz/Dsh with the Baz-aPKC-Par6 complex. We first tested whether Meru is required for the initial asymmetric localization of Baz in SOP cells prior to mitosis. In control SOP cells Baz was asymmetrically enriched at 15 hr APF (*Figure 5A–A'* and *Figure 5—figure supplement 1A–A''*). However, the asymmetric enrichment of Baz was severely impaired in *meru*[1] mutants, so that SOPs were hard to distinguish from surrounding epithelial cells (*Figure 5B–B'* and *Figure 5—figure supplement 1B–B''*). This phenotype is highly reminiscent of the almost complete absence of Baz polarization at interphase in *fz* mutants (*Bellaïche et al., 2001b*; *Besson et al., 2015*). Fz-GFP levels and polarization in *meru*[1] mutants seemed mostly comparable to control nota (*Figure 5J–K'*, *Figure 5—figure supplement 1C*), although Fz-GFP levels appeared reduced in some mutant SOPs. This supports the idea that Meru is required for the initial asymmetric localization of Baz at interphase.

Upon entry into mitosis, the polarization of the Baz-aPKC-Par6 complex is maintained by the antagonism with the opposing Dlg-Pins-Gα$_i$ complex (*Bellaïche et al., 2001b*, *2004*). Interestingly, we found that Baz cortical levels were strongly reduced in metaphase SOPs in *meru* mutants compared to control nota (*Figure 5C–D', G*) and no defined crescent was detectable (*Figure 5C–D', H*) or a very reduced crescent was observed in some cells (*Figure 5—figure supplement 1D–E''*). Similar to loss of *baz* (*Bellaïche et al., 2001b*), Pins remained polarized in *meru* SOPs (*Figure 5E–F', I*). Similarly to Baz, aPKC levels were reduced in mitotic SOPs of *meru* mutants (*Figure 5—figure supplement 1D–E''*), but since aPKC did not interact with Meru in co-IPs (*Figure 5—figure supplement 1F*), this is likely to be an indirect effect due to reduced Baz levels. Thus, Meru is not only important for the initial polarization of Baz, but also needed to enrich Baz and maintain Baz levels during SOP divisions.

## Meru is sufficient to recruit Baz in non-SOP epithelial cells

In order to test whether Meru is sufficient to recruit Baz to Fz/Dsh, we ectopically expressed *GFP-meru* in epithelial cells (non-SOPs) of the pupal notum. Meru and Meru$^{\Delta PBM}$ localized asymmetrically in epithelial cells according to the planar polarization of Dsh (*Figure 6—figure supplement 1A–B''*). Strikingly, wild type Meru but not the Baz binding-deficient Meru$^{\Delta PBM}$ was able to recruit and asymmetrically enrich Baz in non-SOP epithelial cells (*Figure 6A–C* and *Figure 6—figure supplement 1C–D'*). The ability of Meru to bridge the Fz/Dsh-Baz interaction was also seen in co-IP experiments, where Meru, but not Meru$^{\Delta PBM}$, could induce complex formation between Baz and Dsh and strongly promoted the interaction between Baz and Fz (*Figure 6D–E*). The observed binding between Baz and Fz (*Figure 6E* and *Figure 6—figure supplement 1E*) is most likely indirect via Patj (endogenously expressed in S2 cells), as previously described in *Djiane et al. (2005)*. Since Baz mislocalization appears to be a key feature of the *meru* mutant phenotype, we speculated that we might rescue the *meru* defect by increasing Baz dosage. Indeed, ectopic Baz expression in *meru¹* SOPs with *neuralized-Gal4* rescued the *meru¹* stout bristle and missing macrochaete defects (*Figure 6—figure supplement 1F–F'* and *Supplementary file 1*).

# Discussion

## A model for Meru function in ACD

PCP provides the spatial information for the initial polarization of SOPs at interphase, resulting in the planar polarization of Baz, which is uniformly localized prior to SOP differentiation. How Fz/Dsh communicate with Baz and enable its asymmetric enrichment was unknown (*Besson et al., 2015*). Based on our results and previous findings, we propose the following model for the role of Meru in SOP polarization. Upon selection and specification of SOPs, Meru expression is transcriptionally activated by the AS-C transcription factors (*Reeves and Posakony, 2005*). At interphase, planar-polarized Fz/Dsh recruit Meru to the membrane and hence direct its polarization (*Figures 1–3*). Meru in turn positions and asymmetrically enriches Baz, promoting the asymmetry of aPKC-Par6 (*Figures 5–6* and [*Besson et al., 2015*]). Upon entry into mitosis, Meru is also required to retain laterally localized Baz, thus supporting the antagonism between the opposing Dlg-Pins-Gα$_i$ and Baz-aPKC-Par6 complexes, ultimately enabling the correct positioning of cell fate determinants (*Figures 4–5*).

The *meru* mutant cell fate phenotype (bristle duplication or loss) is weaker than the *baz* loss-of-function phenotype, which results in loss of entire SOPs (*Roegiers et al., 2001a*). This is likely due to two factors: (1) unlike *meru* mutants, the full *baz* mutant phenotype is the result of a complete loss of Baz in all cells of the SOP lineage, which is known to cause multiple defects including apoptosis of many sensory organ cells as well as cell fate transformations (*Roegiers et al., 2001a*); (2) since a small amount of Baz is retained at the cortex of some *meru* mutant cells, it is likely that this residual Baz can still be polarized through the antagonistic activity of Pins at metaphase and thus partially rescues SOP polarization (*Bellaïche et al., 2001b*). Indeed, we observed that reduction of *pins* or *baz* levels by RNAi strongly enhanced the *meru* cell specification phenotype (*Figure 3I*). Conversely, supplying excess levels of Baz in a *meru* mutant background presumably restores sufficient Baz at the cortex to rescue the *meru* specification defect, as long as Pins is present to drive asymmetry at mitosis (*Figure 6—figure supplement 1F–F'*).

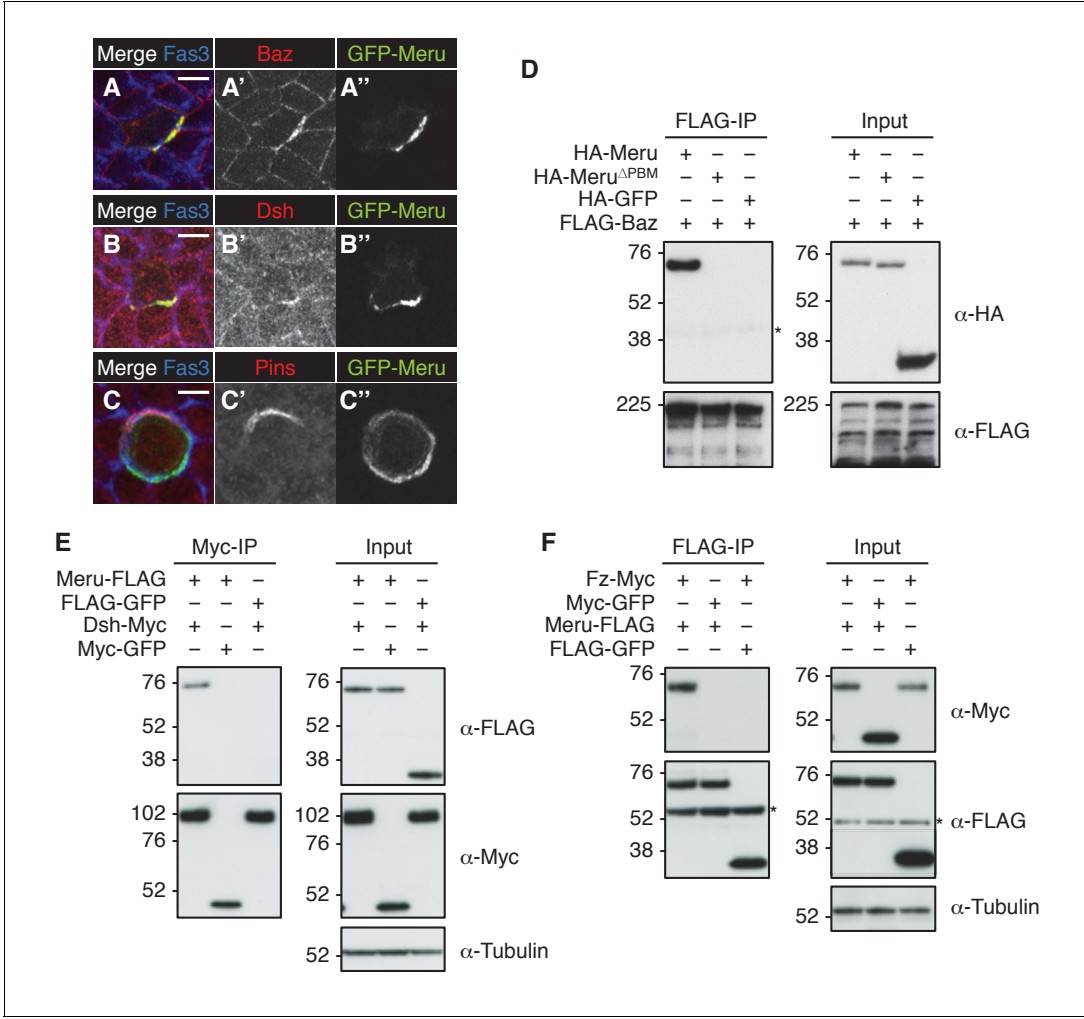

**Figure 2.** Meru co-localizes and associates with posterior SOP polarity components. (**A–A''**) Pupal notum of a *GFP-meru* knock-in animal at 15–16 hr APF stained for Baz (red) and Fas3 (blue). Endogenous GFP-Meru (green) co-localizes with Baz at interphase in SOP cells. (**B–B''**) Pupal notum of a *GFP-meru* knock-in animal at 15–16 hr APF stained for Dsh (red) and Fas3 (blue). GFP-Meru (green) co-localizes with Dsh at interphase in SOP cells. (**C–C''**) Pupal notum of a *GFP-meru* knock-in animal at 15–16 hr APF stained for Pins (red) and Fas3 (blue). GFP-Meru (green) localizes opposite Pins in mitotic SOP cells. (**A–C''**) Scale bar = 5 μm. (**D–F**) Meru co-immunoprecipitates with Baz (D), Dsh (E) and Fz (F) and deletion of the Meru PDZ binding motif (Meru$^{\Delta PBM}$) disrupts the interaction with Baz (D). S2 cells were transfected with the indicated tagged constructs, followed by co-immunoprecipitation and immunoblot analysis. Cross-reacting, non-specific bands are labeled with an asterisk.

The following figure supplement is available for figure 2:

**Figure supplement 1.** Meru co-localizes and associates with SOP polarity components.

While a decrease in cortical Baz can account for the cell specification defects in *meru* mutants, it does not explain the spindle orientation phenotype (*Figure 4E*), since this is not observed in *baz* mutants (*Bellaïche et al., 2001b*; *Roegiers et al., 2001a*). This abnormal spindle alignment could either be due to a decrease in Fz/Dsh levels/activity, or a decrease in the ability of Dsh to recruit the spindle-tethering factor Mud. We could not detect gross abnormalities in Fz levels in *meru* mutants, though the presence of Fz in all neighboring cells would make it difficult to detect subtle decreases in SOPs. Further work will be required to understand Meru's role in spindle orientation.

### Is the function of Meru conserved?

Our analysis of Meru in *Drosophila* is in agreement with the association of human RASSF9 and RASSF10 with both Par3 and PCP proteins we had previously reported (*Hauri et al., 2013*).

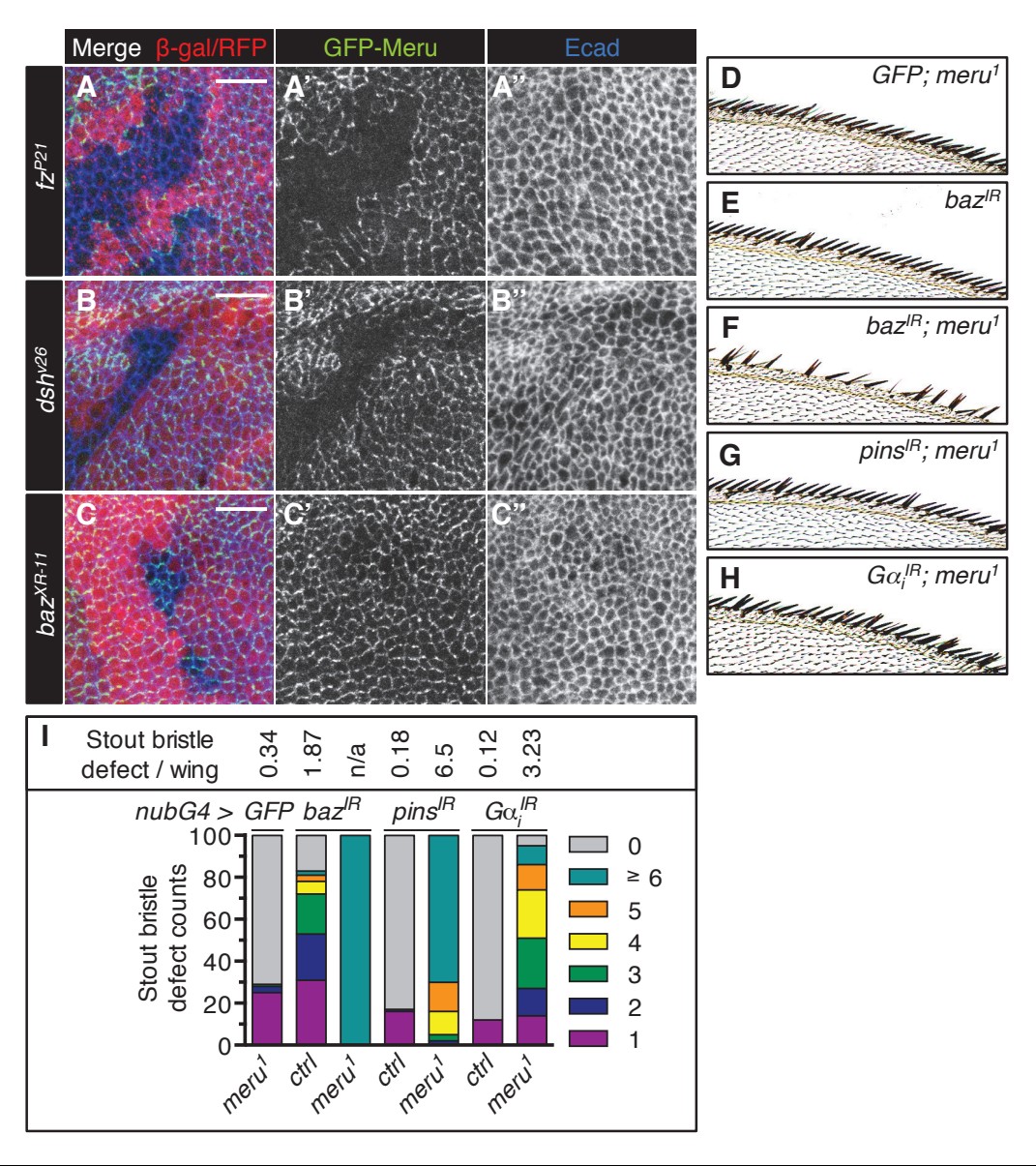

**Figure 3.** Meru cortical localization requires Fz and Dsh and *meru* genetically interacts with different SOP polarity determinants. (**A–C''**) Third instar wing imaginal discs stained for E-cadherin. Mitotic clones were induced with *UbxFlp* for *fz^P21* and *hsFlp* for *dsh^v26* and *baz^XR-11*. Wild type tissue is lacZ positive (stained with anti-β-galactosidase in A-A'') or RFP positive (**B–C''**). *GFP-meru* was expressed under the control of the *Ubi-p63E* promoter. Scale bar = 10 μm. Cortical GFP-Meru disappears in *fz^P21* (**A–A''**) and *dsh^v26* (**B–B''**) clones, but is not affected in *baz^XR-11* (**C–C''**) clones. (**D–H**) Images of the anterior wing margin of adult flies expressing the indicated RNAi lines with *nubbin-Gal4* in wild type (**E**) or *meru^1* mutants. Expression of *GFP* (as a control) in *meru^1* mutants (**D**), *baz^IR* on its own (**E**) and *baz^IR* (**F**), *pins^IR* (**G**) and *Gα_i^IR* (**H**) in combination with *meru^1*. (**I**) Knockdown of *baz*, *pins* or *Gα_i* all cause duplicated stout bristles on their own and strongly increase the defect in *meru^1* mutants (n = 100). See *Supplementary file 1* for statistics.

The following source data and figure supplements are available for figure 3:

**Source data 1.** Source data *Figure 3*.

**Figure supplement 1.** Genetic interactions of *meru* mutants with *dsh* and *fz*.

**Figure supplement 1—source data 1.** Source data *Figure 3—figure supplement 1*.

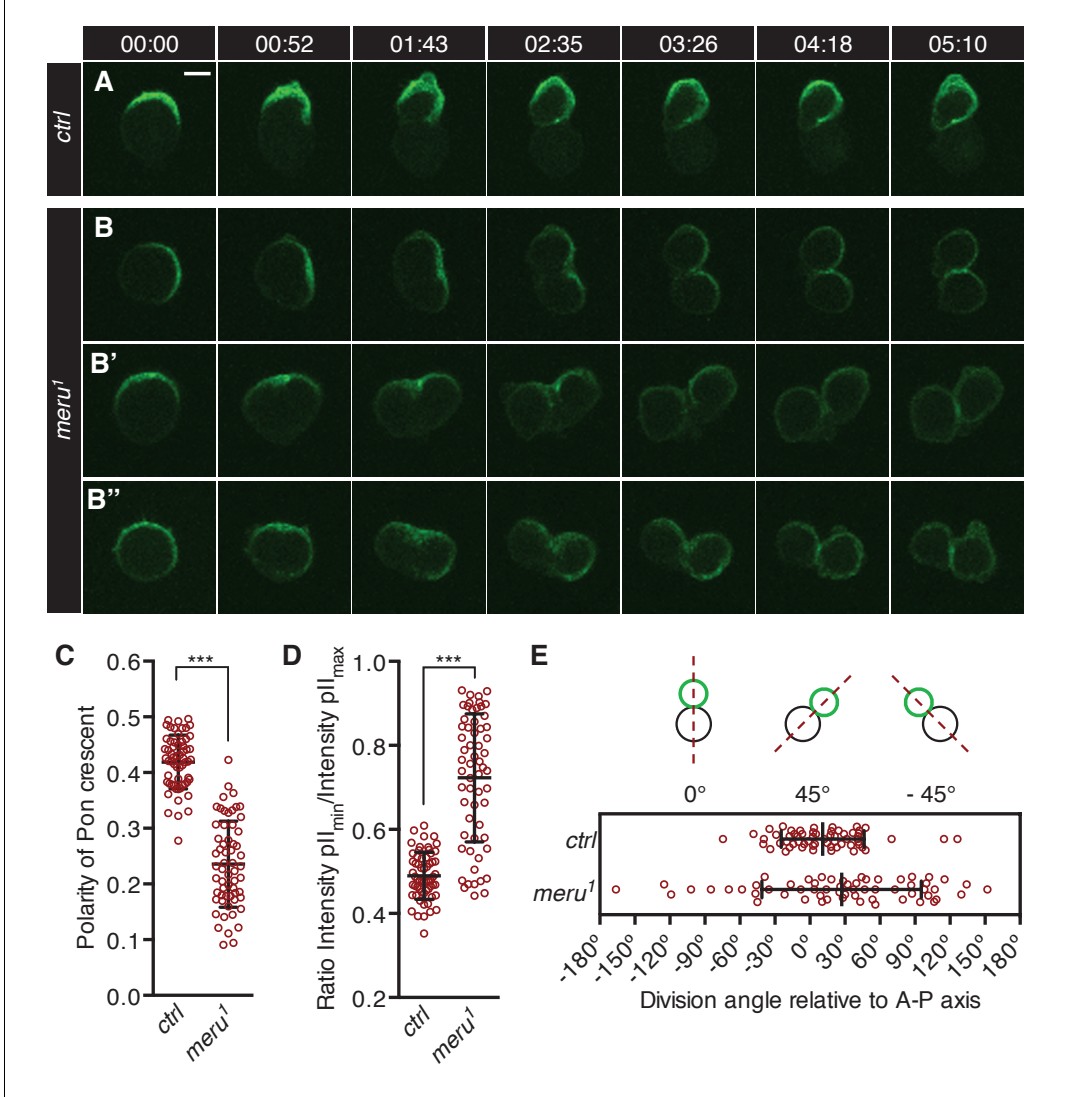

**Figure 4.** *meru* mutants show defects in Pon segregation and spindle orientation. (**A–B''**) Time-lapse analysis of the localization of Pon-GFP during the asymmetric division of a single SOP cell in the pupal notum at 15–17 hr APF. Pon-GFP was expressed under the control of *neuralized-Gal4* and time is given in (min:sec). Scale bar = 5 µm. (**A**) In wild type pupae, Pon-GFP localizes asymmetrically to the anterior side of the pI cell and is distributed into the pIIb cell. (**B–B''**) Examples of SOP divisions in *meru[1]* mutants all resulting in mis-segregation of Pon-GFP into both daughter cells. (**C**) Quantification of the polarization coefficient of the Pon crescent in pI cells (uniform distribution = 0). The Pon crescent is significantly less polarized in *meru[1]* mutants compared to control pupae (Mann-Whitney test: ***p<0.001). (**D**) Relative distribution of Pon-GFP into the two daughter cells (ratios will be approaching one for equal segregations). The Pon segregation is significantly altered in *meru[1]* mutants compared to control pupae (Mann-Whitney test: ***p<0.001). (**E**) The angle of asymmetric SOP divisions relative to the anterior-posterior axis (A-P axis) is randomized in *meru[1]* mutants compared to control pupae. (**C–E**) n = 60 from three pupae for each genotype and error bars represent the mean ± standard deviation.

The following source data is available for figure 4:

**Source data 1.** Source data *Figure 4*.

However, while the interaction with Dsh is conserved between the fly and human proteins, the transmembrane protein Vangl1 (the mammalian homologue of Vang), rather than its antagonist Fz was recovered in the mammalian proteomic analysis (*Figure 2* and [*Hauri et al., 2013*]). This could reflect species-specific differences or altered polarity in the transformed human embryonic kidney 293 cells used for the mammalian work. Although Meru (*CG32150*) was classified as a potential homologue of RASSF10 (*Sherwood et al., 2009*), alignment of the protein sequences showed similar sequence

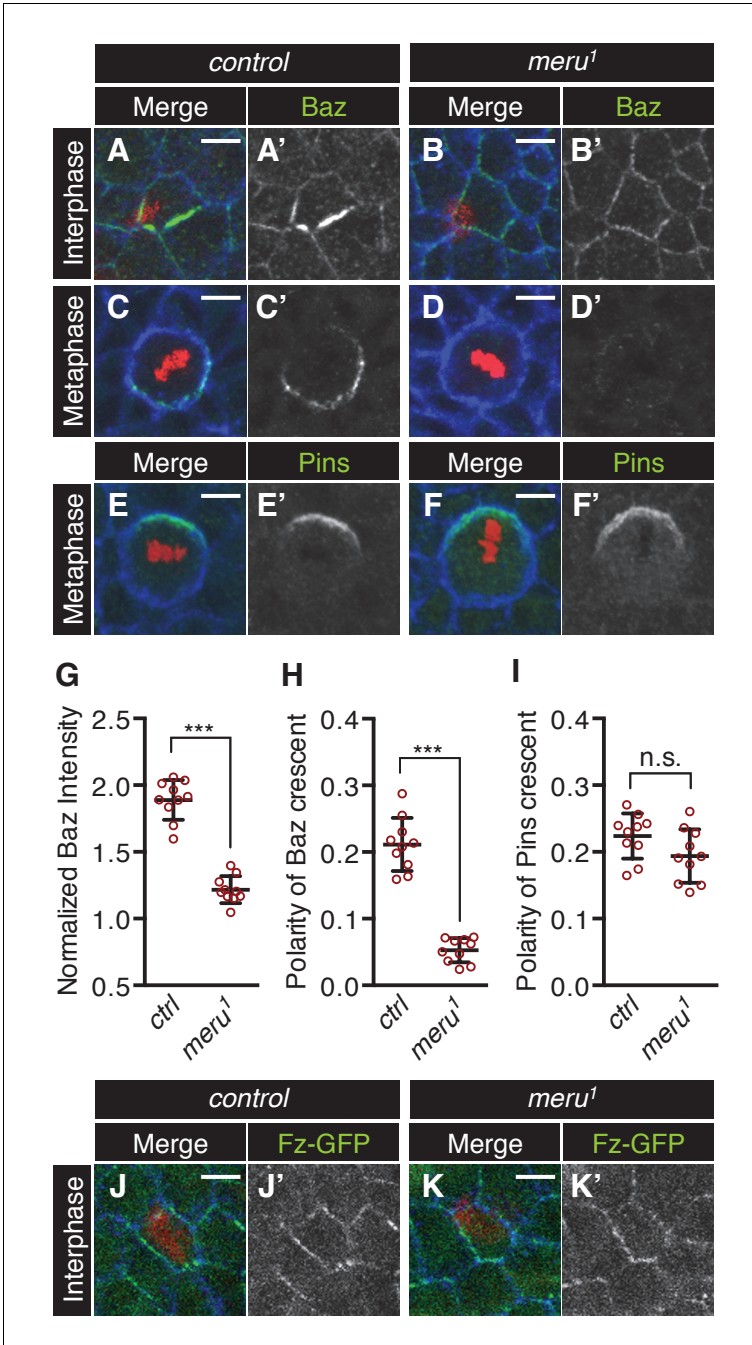

**Figure 5.** The effect of *meru* loss on polarity determinants in SOP cells. (A–D') Pupal nota stained for Baz (green) and E-cadherin (blue) (A-B', 15–16 hr APF) or Fas3 (blue) (C-D', 16–17 hr APF) of control (A–A', C–C') and *meru*[1] (B–B', D–D') pupae (DNA in red). At interphase, Baz appears less asymmetrically enriched in *meru* mutant SOPs (B–B') compared to control (A–A'). In metaphase, loss of *meru* leads to a strong decrease in Baz levels (D–D') compared to control SOPs (C–C'). (E–F') Pupal nota at 16–17 hr APF stained for Pins (green) and Fas3 (blue) of control (E–E') and *meru*[1] (F–F') pupae (DNA in red). Pins is still polarized in *meru* mutant metaphase SOP cells. (G) Quantification of the intensity of the Baz crescent of metaphase SOP cells for control and *meru*[1] nota. The normalized Baz intensity was calculated by dividing the mean grey value intensity at the cell cortex by the mean grey value intensity of the cytoplasm (n = 10). (H–I) Quantification of the polarization coefficient of the Baz crescent (n = 10, same cells as G) (H) or the Pins crescent (n = 10) (I) of metaphase pI cells. While the Baz crescents (H) are significantly less polarized in *meru*[1] mutants compared to control pupae, the Pins crescents are not affected (I). (G–I) Mann-Whitney test: ***p<0.001, n.s. (not significant)=0.0753. Error bars represent the mean ± standard deviation. (J–K') Pupal nota at 15–16 hr APF stained for E-cadherin (blue) of control (J–J') and *meru*[1]

*Figure 5 continued on next page*

*Figure 5 continued*
pupae (**K–K'**) expressing *arm-fz-GFP* (DNA in red). The Fz-GFP signal of interphase SOP cells in *meru[1]* mutants (**K–K'**) is comparable to control pupae (**J–J'**). For quantification, see *Figure 5—figure supplement 1C*.
The following source data and figure supplements are available for figure 5:

**Source data 1.** Source data *Figure 5*.
**Figure supplement 1.** The effect of loss of *meru* on polarity determinants in SOP cells.
**Figure supplement 1—source data 1.** Source data *Figure 5—figure supplement 1*.

identities for both human RASSF9 (31%) and RASSF10 (26%). Thus, further functional work on Meru, its *Drosophila* paralogue *CG13875*, as well as mammalian RASSF9 and RASSF10 is required to understand the evolutionary and functional relationships between these proteins.

Little is known about the in vivo functions of either RASSF9 or RASSF10 in other species. *Xenopus RASSF10* is prominently expressed in the brain and other neural tissues of tadpoles (*Hill et al., 2011*), potentially indicating a function in neurogenesis, a process where ACDs are known to take place. Interestingly, mouse *RASSF9* shows a cell-specific expression in keratinocytes of the skin and loss of *RASSF9* results in differentiation defects of the stratified epidermis (*Lee et al., 2011*). Considering that Par3 is required for ACD of basal layer progenitors of the stratified epidermis (*Williams et al., 2014*) this raises the exciting prospect that RASSF9 might regulate ACD in the mammalian skin.

## Interplay between PCP and apical-basal polarity

The polarization of cells and tissues is essential for their architecture and ultimately allows them to fulfil their function. The polarity machinery can be considered as a series of modules that are combined in a cell or tissue-specific manner, and hence requires specific factors that can create a polarity network appropriate to each tissue and cell type (*Bryant and Mostov, 2008*). We identified Meru as an SOP-specific factor, which is able to link PCP (Fz-Dsh) with apical-basal polarity (Baz). The PCP proteins Vang and Pk promote the positioning of the opposing Dlg-Pins-G$\alpha_i$ complex (*Bellaïche et al., 2004*). Although Vang can directly bind to Dlg, the SOP and neuroblast-specific factor, Banderuola (aka Wide Awake) was recently shown to be required for Dlg localization and could thus constitute a link between the two polarity systems on the opposite side of the cortex (*Bellaïche et al., 2004*; *Mauri et al., 2014*; *Lee et al., 2003*).

There is increasing evidence that cell-type specific rewiring of the polarity modules may be a widespread phenomenon. For instance, in different parts of the embryonic epidermis, Baz is planar polarized by Rho-kinase or by the Fat-PCP pathway (*Marcinkevicius and Zallen, 2013*; *Zallen and Wieschaus, 2004*; *de Matos Simões et al., 2010*), while in the retina, Vang is responsible for Baz polarization (*Aigouy et al., 2016*). Apical-basal polarity can also operate upstream of PCP in some systems, as in *Drosophila* photoreceptor specification, where aPKC restricts Fz activity by inhibitory phosphorylation in a subset of photoreceptor precursors (*Djiane et al., 2005*). Thus, tissue-specific factors are likely to operate in a number of different contexts.

The interplay between PCP and apical-basal polarity is also evident in other species, as Dishevelled has been reported to promote axon differentiation in rat hippocampal neurons by stabilizing aPKC, while *Xenopus* Dishevelled is required for Lethal giant larvae (Lgl) basal localization in the ectoderm (*Dollar et al., 2005*; *Zhang et al., 2007*). Interestingly, both mammalian Par3 and the Vang homologue Vangl2 are required for progenitor cell ACD in the developing mouse neocortex, raising the question as to whether PCP and apical-basal polarity are also connected in mammalian ACDs (*Lake and Sokol, 2009*; *Bultje et al., 2009*). We therefore propose that tissue-specific factors such as Meru might enable the diversity and plasticity observed across different polarized cells and tissues by rewiring the core polarity systems.

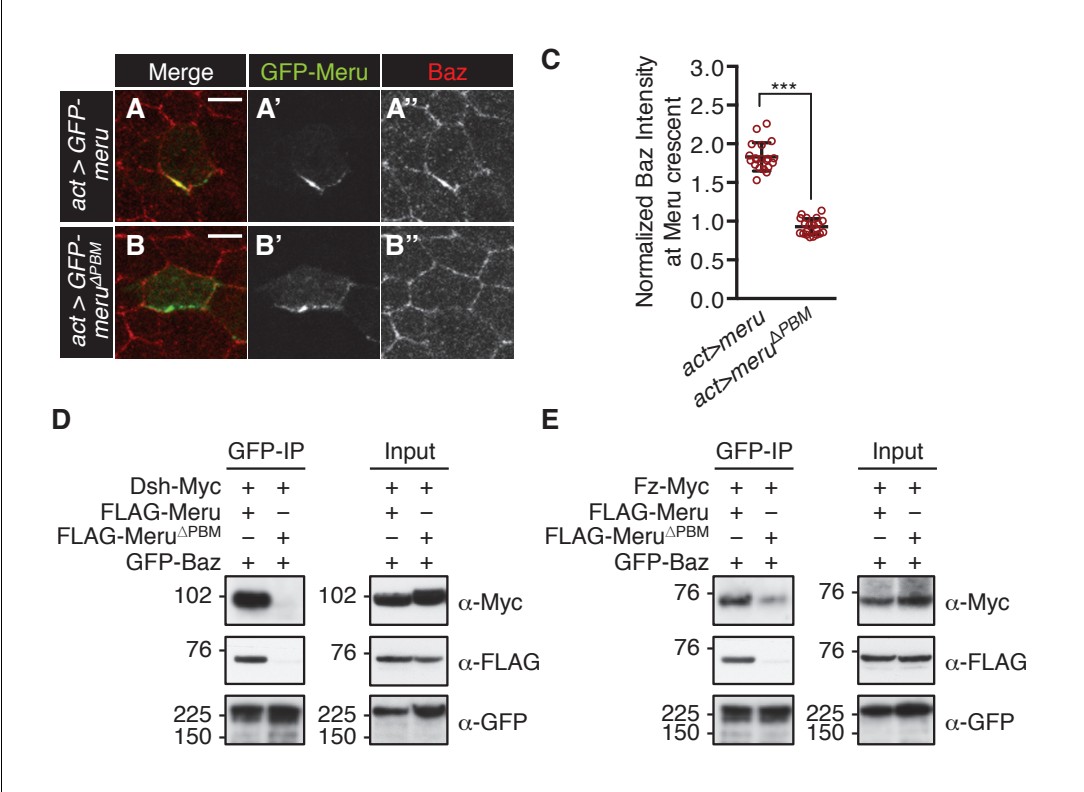

**Figure 6.** Meru has the ability to recruit Baz via its PBM. (**A–B''**) Pupal nota at 15–16 hr APF stained for Baz (red). FLPout clones expressing *GFP-meru* (**A–A''**) or *GFP-meru^ΔPBM* (**B–B''**) under control of *actin-Gal4* were induced with *hsFLP*. GFP-Meru (**A–A''**) can asymmetrically enrich Baz in non-SOP epithelial cells, whereas Baz-binding deficient GFP-Meru^ΔPBM cannot (**B–B''**). Scale bar = 5 μm. (**C**) Quantification of Baz intensity at the Meru (full-length and ΔPBM) crescent. Baz intensities are significantly increased at Meru crescents compared to Meru^ΔPBM crescents (n = 20, Mann-Whitney test: ***p<0.001). Error bars represent the mean ± standard deviation. (**D–E**) Meru can bridge the binding between Baz and Dsh (**D**) and Baz and Fz (**E**) in co-immunoprecipitation experiments. Meru^ΔPBM served as a negative control. S2 cells were transfected with the indicated constructs followed by co-immunoprecipitation and analysis by immunoblot with the indicated antibodies.

The following source data and figure supplements are available for figure 6:

**Source data 1.** Source data *Figure 6*.

**Figure supplement 1.** Meru has the ability to recruit and asymmetrically enrich Baz.

**Figure supplement 1—source data 1.** Source data *Figure 6—figure supplement 1*.

# Materials and methods

## Transgenes and fly stocks

To express GFP-tagged transgenes of *meru* in flies (*ubi-GFP-meru* and *UAS-GFP-meru*), the PhiC31 integrase-mediated system was used. The ORF of *meru* was cloned into the pKC26w_pUbiq_GW vector (identical to pKC26w-pUbiq described in [*Gaspar et al., 2015*; *Zaessinger et al., 2015*] but with addition of an N-terminal GFP-Tag), for expression under the *Ubi-p63E* promoter and into the vector pKC26w_UASt_GW (derived from pKC26w_pUbiq_GW by replacing the *Ubi-p63E* promoter with the *UASt* promoter - from vector pTGW, *Drosophila* Gateway Vector Collection - via MluI/NotI digestion) for Gal4 driver controlled expression. For both vectors a triple stop codon was inserted instead of an ORF to obtain GFP controls. Plasmids were sent to BestGene Inc. for injection (*attP* site: 2L 28E7; genotype: PBac(yellow[+]-attP-3B)VK00002).

The following transgenic fly lines were used: *arm-fz-GFP* (*Strutt, 2001*), *baz-GFP* (FBst0051572, Bloomington *Drosophila* Stock Center), *baz^IR* (5055 R-2, National Institute of Genetics: NIG), *fz^IR*

(17697 R-2, NIG), *Gα$_i$$^{IR}$* (FBst0040890, Bloomington *Drosophila* Stock Center), *neuralized-Gal4* (*Bellaïche et al., 2001a*), *neur-H2B-RFP* (*Gomes et al., 2009*), *pins$^{IR}$* (FBst0053968, Bloomington *Drosophila* Stock Center), *UAS-arm* (gift from J. Vincent), *UAS-baz-GFP* (*Benton and St Johnston, 2003*), *UAS-fz* (FBst0041792, Bloomington *Drosophila* Stock Center), *UAS-Pon-GFP* (*Lu et al., 1999*). The *baz$^{XR-11}$* (*Kuchinke et al., 1998*), *dsh$^{v26}$* (*Klingensmith et al., 1994*) and *fz$^{P21}$* (*Jones et al., 1996*) alleles are all null alleles, whereas *dsh$^1$* is a homozygous viable, PCP-specific allele (*Boutros et al., 1998*).

## Genotypes

*Figure 1* (C–C') *w;; GFP-meru*. (D–D') *w; neur-H2B-RFP/+; GFP-meru*. (E, G) *w;; meru control*. (F, H–H'') *w;; meru$^1$*. *Figure 2* (A–C'') *w;; GFP-meru*. *Figure 3* (A–A'') *yw, UbxFlp/+; ubi-GFP-meru/+; FRT80B arm-lacZ/ FRT80B fz$^{P21}$*. (B–B'') *yw, hsFlp FRT19A, RFP/ FRT19A dsh$^{v26}$; ubi-GFP-meru/+*. (C–C'') *yw, hsFlp FRT19A, RFP/ FRT19A baz$^{XR-11}$; ubi-GFP-meru/+*. *Figure 4* (A) *neuralized-Gal4, UAS-Pon-GFP/+ (control)*. (B–B'') *neuralized-Gal4, UAS-Pon-GFP, meru$^1$/ meru$^1$ (meru$^1$)*. *Figure 5* (A–A', C–C', E–E') *w; neur-H2B-RFP/+*. (B–B', D–D', F–F') *w; neur-H2B-RFP/+; meru$^1$*. (J–J') *w; neur-H2B-RFP/arm-fz-GFP*. (K–K') *w; neur-H2B-RFP/arm-fz-GFP; meru$^1$*. *Figure 6* (A–A'') *yw, hsFlp; act>y$^+$>Gal4, UAS-lacZ/ UAS-GFP-meru*. (B–B'') *yw, hsFlp; act>y$^+$>Gal4, UAS-lacZ/ UAS-GFP-meru$^{ΔPBM}$*. *Figure 1—figure supplement 1* (F–F'') *w;; meru control*. (G–G'') *w;; meru$^1$*. (H–H') *yw, hsFLP/+; FRT80B GFP/ FRT80B meru$^1$*. *Figure 2—figure supplement 1* (B–B'') *baz-GFP/+*. (C–C'') *arm-fz-GFP/+*. (D–E'') *w;; meru control*. *Figure 5—figure supplement 1* (A–A'', D–D'') *w;; meru control*. (B–B'', E–E'') *w;; meru$^1$*. *Figure 6—figure supplement 1* (A–A'') *yw, hsFlp; act>y$^+$>Gal4, UAS-lacZ/ UAS-GFP-meru*. (B–B'') *yw, hsFlp; act>y$^+$>Gal4, UAS-lacZ/ UAS-GFP-meru$^{ΔPBM}$*. (C–C') *UAS-GFP-meru/+; pannier-Gal4/+*. (D–D') *UAS-GFP-meru$^{ΔPBM}$/+; pannier-Gal4/+*.

## Immunofluorescence and live imaging

Third instar larvae or white prepupae were dissected in cold PBS followed by fixation in 4% paraformaldehyde (PFA, Taab) in PBS for 20 min at room temperature. Samples were washed in PBS with 0.1% Triton X-100 (Fisher Scientific) (PBT) four times for 5 min, permeabilised in PBS with 0.3% Triton X-100 for 30 min and blocked in 10% normal goat serum (NGS, MP Biomedicals) in PBT for one hour. Primary antibody incubation in 10% NGS in PBT was carried out overnight at 4°C and was removed in five wash steps in PBT. Secondary antibody incubation was allowed for one hour in 10% NGS in PBT, followed by five wash steps in PBT. For imaging, imaginal discs were mounted in Vectashield anti-fade mounting medium (Vector Laboratories) on glass slides (Thermo Scientific).

For pupal nota stainings, white prepupae were aged for 15–16 hr at 25°C. Nota were dissected in PBS and fixed for 20 min in 4% PFA in PBS. After rinsing with PBT, nota were permeabilised in PBT and blocked in 10% NGS in PBT for one hour each. Primary antibodies were added for two hours and afterwards nota were washed three times for 10 min with PBT. Secondary antibody incubation was allowed for one hour in 10% NGS in PBT and followed by three wash steps with PBT. To allow unfolding of the tissue, nota were incubated in PBS with 50% glycerol overnight at 4°C, and finally mounted in Vectashield with the apical side facing up.

The following primary antibodies were used: mouse anti-Arm 1:10 (N2 7A1, Developmental Studies Hybridoma Bank, DSHB; RRID:AB_528089), rabbit anti-Baz 1:500 (*Walther et al., 2016*), rat anti-Baz 1:500 (*Wodarz et al., 1999*), mouse anti-*β*-galactosidase 1:500 (Z3783, Promega; RRID:AB_430878), rat anti-Dsh 1:1000 and 1:250 (*Shimada et al., 2001*), rat anti-E-cadherin 1:20 (DCAD2, DSHB; RRID:AB_528120), mouse anti-Fas3 1:100 (7G10, DSHB; RRID:AB_528238), mouse anti-Hnt 1:10 (1G9, DSHB; RRID:AB_528278), rabbit anti-Meru 1:500 (this study, Eurogentec), rabbit anti-Pins 1:200 (gift from J. Knoblich) and rat anti-Vang 1:1000 (*Strutt and Strutt, 2008*). Secondary antibodies (1:500) were obtained from Life Technologies and Jackson ImmunoResearch.

Mutant clones in fly tissues were made with the FRT/FLP system using either *UbxFLP* or *hsFLP*, whose expression was induced by heat-shocking larvae twice, 48 and 72 hr after egg laying, for one hour at 37°C. The FLPout system was used to allow the clonal *Gal4* driven expression of *meru* by heat-shocking white prepupae for 9 min at 37°C.

To image ACDs of SOP cells in the pupal notum, pupae expressing fluorophore-tagged proteins were dissected at 15 hr APF. The basal side of the pupa was stuck onto a glass slide with double-sided tape and the pupal case was removed from the head and notum. Two stacks of cover slides

(18 × 18 mm, each stack containing four slides) were positioned in front and behind the pupa. A cover slide (20 × 50 mm) covered in a thin layer of Voltalef Oil 10S (VWR) was carefully placed onto the stacks and pupa, and sealed with nail polish.

Imaging of immunofluorescence and live imaging samples was performed on an SP5 laser scanning confocal (Leica) with a 40x or 63x objective (Z-stack size of 0.5 µm). Images were processed with ImageJ 1.46r (RRID:SCR_003070) and Photoshop CS5.1 (RRID:SCR_014199).

## Antibody generation

A fragment of Meru (amino acids 375–468) with an N-terminal GST-tag was expressed from pGEX-4-T1 (GE Healthcare) in the *E. coli* strain BL21 Codon$^{2+}$ (Promega). Purification of the GST-tagged protein was carried out with Glutathione Sepharose 4B (GE Healthcare). The Meru fragment was then sent to Eurogentec for antibody production in rabbits.

## Analysis and quantification of stainings and time-lapse movies

All analyses and quantifications were carried out with ImageJ 1.46r. For the analysis of Pon (*Figure 4C*), Baz (*Figure 5H*) and Pins (*Figure 5I*) crescent polarity, a freehand line tool was used to draw an outline of each metaphase SOP cell. The outline was straightened and a Plot Profile was created (distance in microns over grey value), with which the polarity of the crescent was determined. We calculated the polarization coefficient (P) according to the following equation:

$$P = \sqrt{a^2 + b^2}$$

With:

$$a = \frac{1}{N}\sum_{k=1}^{N} I_k \cos\left(\frac{k2\pi}{N}\right); \ b = \frac{1}{N}\sum_{k=1}^{N} I_k \sin\left(\frac{k2\pi}{N}\right)$$

$$N = \text{ Number of observations; } I_k = \frac{\text{Grey value } k}{\text{Mean of grey values}}$$

To analyze the polarization of Fz-GFP in interphase SOPs (*Figure 5—figure supplement 1C*), the Plot Profile of the cell outline was determined as described above. The raw grey values were corrected for background noise to obtain the grey values k, which were used to calculate the nematic order parameter S:

$$S = \sqrt{c^2 + d^2}$$

With:

$$c = \frac{1}{N}\sum_{k=1}^{N} I_k \cos\left(2\frac{k2\pi}{N}\right); \ d = \frac{1}{N}\sum_{k=1}^{N} I_k \sin\left(2\frac{k2\pi}{N}\right)$$

$$N = \text{ Number of observations; } I_k = \frac{\text{Grey value } k}{\text{Mean of grey values}}$$

For the analysis of the Pon-GFP distribution from the mother SOP cell into daughter cells (*Figure 4D*) the Pon-GFP signal's grey value was measured for both daughter cells (pII$_{min}$ and pII$_{max}$), and the Pon segregation was calculated by dividing the smaller grey value (intensity pII$_{min}$) by the bigger grey value (intensity pII$_{max}$).

The division angle relative to the anterior-posterior body axis (A-P axis: 0°) was measured as illustrated in *Figure 4E*. Clockwise divisions were given positive values (0 to 180°), and counter clockwise divisions negative values (0 to −180°).

Normalized Baz intensities were determined in *Figure 5G* by dividing the mean of the grey values of the entire cell outline (freehand line tool) by the mean of the grey values of a sample rectangle in the cytoplasm.

Normalized Baz intensities were calculated in a different way for *Figure 6C* in order to compare the Meru-containing cortex to the non-Meru containing cortex. The mean of the grey values of the

Baz staining along the Meru crescent (freehand line tool) was divided by the mean of the grey values along the remaining membrane (freehand line tool).

## Analysis of mechanoreceptor defects in adult flies

Duplications of stout bristles of adult wings were quantified for 100 wings (the left and right wing of 50 flies) of each genotype on a Leica dissection microscope and the same flies were also checked for missing thoracic macrochaetes. For imaging, adult wings were mounted in Euparal (ALS) and dried at 65°C overnight. Images were acquired with a Zeiss Axioplan2 microscope and a Leica DFC420 camera. Images of the notum and abdomen of adult flies were taken with a Leica MZ7.5 dissection microscope and a Leica DFC420 camera.

## Statistical analysis

To test whether mean values differed significantly, an unpaired nonparametric t-test (Mann-Whitney test) was performed with the Prism 6 software (RRID:SCR_002798). Fisher's exact test (Freeman-Halton extension) (*Freeman and Halton, 1951*) with the SPSS 21 software (RRID:SCR_002865) was used for the bristle (margin and thoracic) analysis to test for significant differences in the distributions of bristle defect categories between different genotypes. Error bars represent the mean ± the standard deviation in all figures.

## S2 cells, co-immunoprecipitation, western blotting

S2 cells (RRID:CVCL_Z232) were maintained in Schneider's *Drosophila* Medium (Gibco) with 10% Fetal Bovine Serum (Sigma-Aldrich) and 1% Penicillin-Streptomycin (10000 U/mL, Gibco). To express tagged proteins in S2 cells, ORFs were cloned from fly mRNA or cDNA from the *Drosophila* Genomics Resource Center (https://dgrc.bio.indiana.edu/Home) into vectors of the *Drosophila* Gateway Vector Collection. S2 cells were transiently transfected using the Effectene Transfection Reagent (Qiagen). dsRNA treatment of cells prior to transfection was carried out as previously described (*Aerne et al., 2015*). Cells were lysed in 200 µL of lysis buffer (50 mM Tris HCl pH 8, 150 mM NaCl, 1% (v/v) IGEPAL (CA-630), 1 mM EGTA, 100 µL/mL 0.5 M NaF, 10 µL/mL phosphatase inhibitor cocktail 2 (Sigma-Aldrich), 10 µL/mL phosphatase inhibitor cocktail 3 (Sigma-Aldrich), protease inhibitor cocktail (Roche)). For co-IP experiments, cleared cell lysates were added to Anti-FLAG M2 Affinity Gel (Sigma-Aldrich; RRID:AB_10063035), to GFP-Trap_A beads (ChromoTek) or to anti-Myc antibody bound Protein A Sepharose 4B Fast Flow (Sigma) and incubated for 1.5 hr at 4°C. Input and co-IP samples were analyzed by SDS-PAGE and western blot. The following primary antibodies were used for western blotting: mouse anti-FLAG 1:1000 (M2, Sigma-Aldrich; RRID:AB_259529) rabbit anti-FLAG 1:1000 (F2555, Sigma-Aldrich; RRID:AB_796202), mouse anti-GFP 1:1000 (3E1, in house), mouse anti-HA 1:5000 (12CA5, CRUK), mouse anti-Myc 1:1000 (sc-40, Santa Cruz; RRID:AB_627268), rabbit anti-Myc 1:1000 (sc-798, Santa Cruz; RRID:AB_631274) and mouse anti-Tubulin 1:2000 (E7, DSHB; RRID:AB_528499). HRP coupled secondary antibodies derived from GE Healthcare and were diluted 1:5000.

## Genomic engineering

Deletion mutants of *meru* as well as a N-terminal GFP knock-in were made with the CRISPR/Cas9 system.

To generate deletion mutants of *meru* gRNA pairs were used (see *Figure 1—figure supplement 1A*). gRNAs were designed using the Perrimon lab's website (http://www.flyrnai.org/crispr2/). gRNA target sequences (5′−3′): gRNA 1a: CCTCTTAATCGATCTACATACTC, gRNA 1b: CCAACTGTATAGGGGTACCGAAA, gRNA 2a: GGCCCACAGGGGCCGTGAAATGG, gRNA 2b: CCTCTATGGCGTTAATAGCACTG. For the GFP knock-in the gRNA target site was chosen close to the ATG to facilitate the integration of the homologous recombination construct. The gRNA target sequence was CCCAACAACAGAACTCCACATTC. gRNA expression plasmids (pCFD3-dU6:3gRNA, Addgene) for injection into flies were prepared as described in the cloning protocol provided by the Bullock lab's website (http://www.crisprflydesign.org/grna-expression-vectors/). The *meru* homology region for the GFP knock-in was cloned into the pCR2.1 vector (Life Technologies) using the following primers: CGTTCAAGGATATCTAGTGGCAGG, CGGATTATTGCCGCAGTAGAATCC. To prevent potential cutting of the homologous recombination construct, a silent mutation was introduced at the gRNA

target site using the primer CGGTCAATGGAATATGGCGCCACAACAACAGAACTCCACATTC. The eGFP coding sequence was inserted at the *meru* ATG using overlap extension PCR cloning (*Bryksin and Matsumura, 2010*). The primers to generate the megaprimer were: GG TTTTTCAAAAGGCGGTCAATGGAATATGGTGAGCAAGGGCGAGGAGC, GGAGTTCTGTTGTTG TGGCGCCATCGTGGACCGGTGCTTGT. Plasmids were confirmed by sequencing and then sent for injections to the Fly Facility (Department of Genetics) at the University of Cambridge. The gRNA expression plasmid pairs for the deletion mutants (250 ng/µL of each plasmid) were injected into embryos of a germline restricted nos-Cas9 line: $y^1$, *M(nos-Cas9.P)ZH-2A, w\** (FBst0054591). The gRNA and homologous recombination plasmids were combined at 100 ng/µL and 500 ng/µL respectively.

In the case of the deletion mutants, surviving founders were crossed to a balancer stock (*yw;;Dr/TM3*). Their progeny were crossed to a *meru* deficiency line (*Df(3L)BSC575/TM6B*) to identify mutants by phenotype and genotyped by PCR. Males positive for a deletion were crossed to a balancer stock (*w;;TM3/TM6B*) to establish stable lines balanced over *TM6B*. From these lines, DNA was extracted and the *meru* locus sequenced to characterize the deletion. To obtain a control stock with an identical genetic background (*meru control*), exactly the same crosses were made, starting with a non-injected male of the *nos-Cas9* line. For the GFP knock-in, founders were crossed to a balancer stock (*w;;TM3/TM6B*). Male progeny were crossed to the same balancer stock and genotyped by PCR. The progeny of single males positive for the knock-in were then used to establish stable lines balanced over *TM6B*.

## Acknowledgements

We would like to thank B Baum, Y Bellaïche, M Bienz, M Gho, J Knoblich, F Pichaud, D Strutt, T Uemura, J Vincent, A Wodarz, the Bloomington *Drosophila* Stock Center, the *Drosophila* Genomics Resource Center and the Developmental Studies Hybridoma Bank for flies and/or reagents. We are very grateful to N Rodriguez and A Ainslie for advice on notal stainings and time-lapse microscopy, to J R Davis for advice on image and statistical analysis, to G Salbreux for advice on data analysis, X Tapon for help with data analysis, and L Kester for cloning *meru*. We thank F Schweisguth, Y Bellaïche, D Strutt, A Chalmers and Tapon lab members for advice and F Schweisguth and Tapon lab members for comments on the manuscript. We are grateful to the Crick light microscopy and fly facilities. This work was supported by the Francis Crick Institute, which receives its core funding from Cancer Research UK (FC001175), the UK Medical Research Council (FC001175), and the Wellcome Trust (FC001175), as well as a Wellcome Trust Investigator award (107885/Z/15/Z). Research in the Gstaiger lab is supported by the European Union 7th Framework Programme SYBILLA (Systems Biology of T-Cell Activation) and the Innovative Medicines Initiative project ULTRA- DD (grant agreement n° 115766).

## Additional information

### Funding

| Funder | Grant reference number | Author |
|---|---|---|
| Francis Crick Institute | FC001175 | Jennifer J Banerjee<br>Birgit L Aerne<br>Maxine V Holder<br>Nicolas Tapon |
| Wellcome | 107885/Z/15/Z | Jennifer J Banerjee<br>Birgit L Aerne<br>Maxine V Holder<br>Nicolas Tapon |
| Innovative Medicines Initiative project ULTRA-DD | 115766 | Simon Hauri<br>Matthias Gstaiger |
| European Union 7th Framework programme | SYBILLA | Simon Hauri<br>Matthias Gstaiger |

The funders had no role in study design, data collection and interpretation, or the decision to submit the work for publication.

## Author contributions
JJB, Conceptualization, Investigation, Writing—original draft, Writing—review and editing; BLA, MVH, Investigation, Writing—review and editing; SH, MG, Conceptualization, Resources; NT, Conceptualization, Supervision, Funding acquisition, Writing—original draft, Writing—review and editing

## Author ORCIDs
Nicolas Tapon, http://orcid.org/0000-0001-5267-6510

## Additional files

### Supplementary files
• Supplementary file 1. Test of significance for mechanoreceptor defects between genotypes using Fisher's exact test (Freeman-Halton extension) (related to *Figures 1,3* and *Figure 3—figure supplement 1*; *Figure 6—figure supplement 1*).

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
