## [Decision Letter]

Thank you for submitting your article "Meru couples planar cell polarity with apical-basal polarity during asymmetric cell division" for consideration by *eLife*. Your article has been reviewed by two peer reviewers, and the evaluation has been overseen by a Reviewing Editor and Fiona Watt as the Senior Editor. The following individual involved in review of your submission has agreed to reveal his identity: Jeffrey D Axelrod (Reviewer #3).

The reviewers have discussed the reviews with one another and the Reviewing Editor has drafted this decision to help you prepare a revised submission.

This paper addresses the role of Meru, a RASSF9/10 homologue, in *Drosophila* sensory organ precursor (SOP) polarization that is critical for sensory organ development and orientation. Polarity is a feature common to all epithelial cells, and A/B polarity often co-exists with planar cell polarity (PCP). The latter orients the cells, and cellular features like spindle orientation, in the plane of the epithelial sheet. PCP is established by asymmetric localization of the two complexes of the core Frizzled (Fz)/PCP factors, which also include Dsh as a binding partner of Fz. The authors show in this paper that Meru is a factor that participates in linking the Fz/Dsh PCP complex to Bazooka (Baz/Par3) in *Drosophila* SOPs. They suggest that the core PCP components initiate planar polarization of apical-basal determinants, ensuring asymmetric division into daughter cells of different fates, and that Meru, is expressed specifically in SOPs and recruited to the posterior cortex by Fz/Dsh complexes. Meru in turn polarizes the apical-basal polarity factor Baz/Par3. They propose that Meru belongs to a class of cell or tissue-specific proteins that act to modulate the read-outs core PCP machinery.

The reviewers considered the work to be of high quality and broad interest and recommend publication subject to the minor revisions listed below.

Figure 1: it is nicely shown that Fz and Dsh can bind Meru (panels E, F) and that Meru binds Baz/Par3 (panel D). What remains unclear is whether Baz can also bind Fz. This is relevant, as it was shown earlier that the Fz C-tail binds Baz, see Djiane et al. 2005, in the *Drosophila* eye disc during PCP establishment. In the eye, Baz is part of the Fz-Dsh complex to protect Fz from aPKC phosphorylation. The authors should check whether Baz and Fz can also interact in the notum and check if a similar relationship exists, or at least discuss the issue.

Figure 3: It would help if high magnification panels could be shown to better represent the staining show in panels A-C.

Text issues:i) The dsh-1 allele was first described by Boutros et al. 1998 (before the paper by Axelrod et al. 1998).

ii) The authors use the gene name Strabismus/Stbm, which is one of the two *Drosophila* names for Van Gogh (Vang)/Strabismus. As the vertebrate community has adopted the "Van Gogh-like" nomenclature, I suggest the authors also call the gene at least Vang/Stbm, or just Vang, to keep the commonly accepted gene name across species, and thus improve clarity for non *Drosophila* researchers.

Figure 5: In panels J-K, the Fz protein asymmetry/polarization is compared in wt vs. mutant backgrounds, and it is concluded that Fz polarization is not affected in Meru mutants. However, to make this statement this should be quantified. Such asymmetries can and should be quantified with the Packing Analyzer software available for free from the Eaton lab and Benoit Aigouy. This is now standard in the field.

---

## [Author Response]

*[…] The reviewers considered the work to be of high quality and broad interest and recommend publication subject to the minor revisions listed below.*

*Figure 1: it is nicely shown that Fz and Dsh can bind Meru (panels E, F) and that Meru binds Baz/Par3 (panel D). What remains unclear is whether Baz can also bind Fz. This is relevant, as it was shown earlier that the Fz C-tail binds Baz, see Djiane et al. 2005, in the Drosophila eye disc during PCP establishment. In the eye, Baz is part of the Fz-Dsh complex to protect Fz from aPKC phosphorylation. The authors should check whether Baz and Fz can also interact in the notum and check if a similar relationship exists, or at least discuss the issue.*

The paper Djiane et al. 2005 shows that the C-terminus of Fz can bind to Patj, but not to Baz in a yeast two-hybrid experiment (Figure 3). This interaction is mediated by the 4^th^ PDZ domain in Patj (Figure 3), as shown by GST pull-downs. Based on these results, the authors conclude that Fz binds directly to Patj. They further show in co-IPs of Fz overexpression in S2 cells, that endogenous Patj, aPKC and Baz can be recovered suggesting that these four proteins can form a complex (most likely via the Patj/Fz association). In vivo, Djiane et al. find that Baz levels are enriched in the Fz positive R3/R4 photoreceptors. The authors suggest that Baz inhibits aPKC/Patj in these cells and thus supports Fz activity. The paper does not, however, suggest the existence of Baz/Fz/Dsh complexes, nor does it show any biochemical association between these proteins in vivo.

Nevertheless, we did consider the possibility that Patj in SOPs might participate in Baz recruitment to Fz/Dsh. However, this is unlikely, as a recent publication by Perez-Mockus et al. 2017, demonstrates that Neuralized can reduce Patj levels (most likely indirectly by reducing Stardust levels). This suggests that Patj levels are likely reduced in SOPs due to high Neuralized levels and therefore unlikely to mediate Baz polarization in these cells. In addition, like Baz and Fz/Dsh, Patj is expressed in neighboring epithelial cells, where Baz is not planar polarized unless Meru is ectopically expressed. Thus, it is very unlikely that Patj is involved in Baz planar polarization in SOPs.

In our manuscript, we show that Meru strongly enhances the binding between Fz and Baz (Figure 6). In order to address the reviewer’s concerns, we performed co-IPs in S2 cells with extra controls and confirmed the result that Baz and Fz co-immunoprecipitate (as seen in Djiane et al. 2005), likely via endogenous Patj. We have added these data to Figure 6—figure supplement 1. However, as shown in Figure 6, this interaction is strongly boosted by addition of Meru. We added the following paragraph to our Results section:

The observed binding between Baz and Fz (Figure 6 and Figure 6—figure supplement 1) is most likely indirect via Patj (endogenously expressed in S2 cells), as previously described in (Djiane et al., 2005).

*Figure 3: It would help if high magnification panels could be shown to better represent the staining show in panels A-C.*

As requested, we performed new stainings and used a higher magnification objective to create higher resolution images. We replaced panels A-C in Figure 3 with these new ones and added higher magnification panels (A-C) to Figure 3—figure supplement 1.

*Text issues:i) The dsh-1 allele was first described by Boutros et al. 1998 (before the paper by Axelrod et al. 1998).*

We apologize for citing the wrong reference and have updated the reference for the *dsh-1* allele to Boutros et al. 1998.

*ii) The authors use the gene name Strabismus/Stbm, which is one of the two Drosophila names for Van Gogh (Vang)/Strabismus. As the vertebrate community has adopted the "Van Gogh-like" nomenclature, I suggest the authors also call the gene at least Vang/Stbm, or just Vang, to keep the commonly accepted gene name across species, and thus improve clarity for non Drosophila researchers.*

As requested, we replaced Stbm by Vang in the text and Figure 2—figure supplement 1’.

*Figure 5: In panels J-K, the Fz protein asymmetry/polarization is compared in wt vs. mutant backgrounds, and it is concluded that Fz polarization is not affected in Meru mutants. However, to make this statement this should be quantified. Such asymmetries can and should be quantified with the Packing Analyzer software available for free from the Eaton lab and Benoit Aigouy. This is now standard in the field.*

We agree with the reviewers that quantification of Fz polarization would be necessary to draw this conclusion. Therefore, we took images of control and *meru* mutant pupae expressing *arm-fz-GFP/ neur-H2B-RFP* and quantified Fz polarization in interphase SOP cells by calculating the nematic order parameter S (see Sagner et al. 2012). We added the following to the Materials and methods section:

To analyze the polarization of Fz-GFP in interphase SOPs (Figure 5—figure supplement 1) the Plot Profile of the cell outline was determined as described above. The raw grey values were corrected for background noise to obtain the grey values k, which were used to calculate the nematic order parameter S:

With:

This analysis confirmed that there is no detectable difference in Fz polarization between wild type and meru mutant animals. We note that the nematic order of Fz-GFP seems low in the pupal notum compared, for example, to the late pupal wing.

We added this result to Figure 5—figure supplement 1 (C).